# Manufacturing of a Transdermal Patch in 3D Printing

**DOI:** 10.3390/mi13122190

**Published:** 2022-12-10

**Authors:** Isabella Villota, Paulo César Calvo, Oscar Iván Campo, Luis Jesús Villarreal-Gómez, Faruk Fonthal

**Affiliations:** 1Biomedical Engineering Research Group—GBIO, Universidad Autónoma de Occidente, Cali 760030, Colombia; 2Facultad de Ciencias de la Ingeniería y Tecnología, Universidad Autónoma de baja California, Tijuana 21500, Baja California, Mexico; 3Science and Engineering of Materials Research Group-GCIM, Universidad Autónoma de Occidente, Cali 760030, Colombia

**Keywords:** microneedles, transdermal drug delivery, 3D printing, finite element analysis

## Abstract

Diabetes mellitus is an endocrine disorder that affects glucose metabolism, making the body unable to effectively use the insulin it produces. Transdermal drug delivery (TDD) has attracted strong interest from researchers, as it allows minimally invasive and painless insulin administration, showing advantages over conventional delivery methods. Systems composed of microneedles (MNs) assembled in a transdermal patch provide a unique route of administration, which is innovative with promising results. This paper presents the design of a transdermal patch composed of 25 microneedles manufactured with 3D printing by stereolithography with a class 1 biocompatible resin and a printing angle of 0°. Finite element analysis with ANSYS software is used to obtain the mechanical behavior of the microneedle (MN). The values obtained through the analysis were: a Von Misses stress of 18.057 MPa, a maximum deformation of $2.179\times 10^{-3}$, and a safety factor of 4. Following this, through a flow simulation, we find that a pressure of 1.084 Pa and a fluid velocity of 4.800 $\frac{m}{s}$ were necessary to ensure a volumetric flow magnitude of $4.447\times 10^{-5}\frac{{\mathrm{cm}}^{3}}{s}$. Furthermore, the parameters found in this work are of great importance for the future implementation of a transdermal drug delivery device.

## 1. Introduction

Diabetes mellitus is an endocrine disorder that affects glucose metabolism, making the body unable to use the insulin it produces effectively. Over time, this disorder can cause severe damage to organs and systems [1,2,3,4]. 

The World Health Organization (WHO) reports that approximately 62 million people in the Americas (422 million people worldwide) have diabetes—the majority living in low- and middle-income countries—and 244,084 deaths (1.5 million worldwide) are directly attributed to diabetes yearly. The number of cases and the prevalence of diabetes have increased steadily over the last few decades [5].

According to the Pan American Health Organization (PAHO), diabetes is one of the main causes of chronic problems such as kidney failure, blindness, heart attacks, strokes, and amputation of lower limbs. Therefore, it is a considerable public health problem that many researchers focus on to find solutions or improvements to diabetes treatments [2].

Conventional drug delivery methods have different limitations due to the physicochemical properties of the drug and the effects it may cause after application [6]. Furthermore, traditional insulin injections use hypodermic needles, which is a quick and easy way to administer the drug but generates discomfort and pain at the time of application [7]. Therefore, new approaches have been developed to improve this form of application.

Transdermal drug delivery (TDD) has attracted strong interest from researchers, as it allows minimally invasive and painless insulin administration, showing advantages over conventional delivery methods [8,9,10,11].

Microneedles represent a physical enhancement method for transdermal and intradermal delivery of drugs [1], which is innovative and with promising results [12,13,14,15,16,17,18,19,20]. The relationships between design and manufacturing parameters and quality and performance can be systematically explored on 3D-printed microneedles (MNs) [21,22]. MNs have lengths from 150 μm to 1500 μm—this being a sufficient length to release the drug into the epidermis—and widths between 50 μm and 250 μm [20]. MNs penetrate the skin without stimulating the dermal nerves; they are large enough to penetrate the epidermis and stratum corneum without producing pain [18,19]. The printing resolution and material properties are the two critical parameters that significantly affect the outcome of microneedle printing endeavors. In this way, [23] showed a microneedle concept made by 3D printing using a bio-compatible resin of class 1 as a promising option for transdermal drug-delivery devices.

On the other side, this MN patch fabrication technique is a valuable contribution to microneedle manufacture techniques since conventional techniques, such as micro-milling and injection molding, among others, have some cost-saving limitations [24]. Meanwhile, stereolithography (SLA)-printed patches present important characteristics, such as low printer costs, printing inks, and fast manufacturing times [14], allowing this technique to make considerable contributions to the manufacture of devices for transdermal drug delivery. 

In 2010, two research groups worked on the two-photon polymerization technique. Gittard et al. discuss two-photon polymerization, a laser-based rapid prototyping technique, and describe how this technique can be used to make microneedles with exceptional mechanical properties and discusses the potential applications of microneedles in drug delivery [25]. Doraiswamy et al. discuss the fabrication of microneedles using two-photon polymerizations and show that these microneedles can be used for the transdermal delivery of nanoscale pharmacologic agents [26]. 

Five years later, Lu et al. presented the fabrication of drug-loaded microneedle arrays using micro stereolithography. These arrays are poly (propylene fumarate) mixed with diethyl fumarate to control viscosity and improve mechanical properties [27]. Dardano et al. show how MNs from 100 µm to 2 mm can be obtained by changing the exposure time and power density during photolithography. In addition, different MN shapes, such as cylindrical, conic, or lancet-like, can be achieved for specific applications such as micro-indentation or drug delivery [28]. Ali et al. propose a microfabrication process for polymer MNs, using DLP-based projection-based stereolithography. The fabrication is performed with continuous movement of the platform in the vertical direction. The results indicate that polymer MNs with appropriate geometry can be fabricated using this technique [29].

In the last decade, various research groups presented computer-aided design (CAD) as an important tool for fabrication. Ge et al. presented the different advantages and challenges that arise in micromanufacturing with 3D printing. In their study of hydrogels, they show the difficulties that occur with materials manufactured at micrometer dimensions; they further show the importance of computational studies using models and simulations to better understand the manufacturing process [30]. Recently, Yang et al. mentioned that high precision and accuracy enabled the SLA technique to fabricate hollow microneedles capable of transdermal drug delivery [31], and Xenikakis et al. reported a typical solid transdermal microneedle array using a polymer-based material [32]. They utilized finite element analysis (FEA) to simulate the printed microneedle array insertion.

Micro-modeling techniques are used in most cases to manufacture MNs arrays for drug delivery [33,34,35,36]; however, other applications of micro-modeling approaches include the 3D reconstruction of biological tissues [37,38], numerical simulation (e.g., recovery of gas hydrates) [39], computational platform (e.g., finite element for heart ventricles) [40], finite element analysis of rubberized concrete interlocking masonry [41], among others. Other methods that have been explored for manufacturing include 3D printing, which has allowed innovation in different fields, such as pharmaceutical and biomedical sciences, among others [42].

Although 3D printing by FDM using PLA has been used to manufacture microneedles [43], they can be further degraded using potassium hydroxide to obtain a thinner diameter. However, such a process does not allow adequate control of the shape and design of the microneedle; thus, 3D printing methods, such as SLA, have been preferred to allow higher resolution and lower cost [44].

The manufacture of microneedle patches for TDD by 3D printing, a relatively recent method, presents challenges that must be overcome, such as the sensitivity to different parameters in the pre-printing (degradation of resin, geometric parameters) and post-printing (type of solvent used for cleaning, UV exposure time, thermal post-cure) process, for printing quality [45,46].

Microneedles represent one of the microscale physical enhancement methods that greatly expand the spectrum of drugs for transdermal and intradermal delivery. Researchers suggest that 3D printing can manufacture microneedles and that the printing resolution and material properties are important considerations. However, there remain challenges with sustained delivery, efficacy, cost-effective fabrication, and large-scale manufacturing [1,21,22,23]. These gaps in microneedle manufacturing technologies represent a challenge for 3D printing.

This paper presents the design of a patch intended for use in transdermal drug delivery using computer-aided design (CAD).

The geometry of a single-bevel hypodermic needle inspires the MNs in this patch. The design of this microneedle was mechanically validated using finite element analysis (FEA). A computational fluid dynamic (CFD) configuration simulation was performed to obtain the necessary parameters for the future design of the TDD device that will contain this designed patch. In addition, the patch was manufactured by 3D printing with a class 1 biocompatible resin on the Form2-Formlabs printer.

## 2. Materials and Methods

### 2.1. Materials

The material used to manufacture the MN patch was surgical Guide class 1 biocompatible resin (Formlabs). The resin is non-cytotoxic, non-sensitizing, and non-irritating and complies with ISO 10993-1:2018 [47]. The surgical guide requires post-curing to achieve biocompatibility and optimal mechanical properties. In addition, this material is autoclavable, an important feature for medical uses.

Furthermore, 99% isopropyl alcohol was used to wash the printed parts, as indicated by the resin manufacturer’s specifications.

### 2.2. Computer-Aided Design (CAD) of MN

The structural design of the MN was modeled with Solidworks 2019 software and inspired by the one-plane bevel-tipped needle design. The MN had a length of 450 μm, an inner diameter (β) of 100 μm, an outer diameter (α) of 232 μm, and a tip angle (θ) of 45° between the lateral surface and the beveled plane, see Figure 1. This MN was evaluated in detail in [23].

This relatively simple MN structure allows an adequate load distribution over the structure, allowing the penetration of the MN into the skin without structural damage.

### 2.3. Finite Element Analysis of MN

The mechanical analysis of the MN made it possible to observe its behavior when inserted into the skin; this was simulated when the MN was exposed to the force necessary to penetrate the skin. That is important because the MN must correctly withstand the skin’s resistance without generating adverse effects such as buckling during insertion.

To improve the simulation response, it was necessary to know the tensile strength and Young’s modulus of the surgical guide resin, which were 73 MPa and 2.9 GPa, respectively. In addition, a force of 0.5 N was used to simulate the force needed to pierce the skin in a hydrated condition [23,48].

#### Fluid Simulation in MN

The MNs are responsible for administering the drug transdermally, i.e., the fluid entering the epidermis circulates through the MNs. Therefore, it is important to know the mass flow, volumetric flow, and fluid velocity of the MN for a future drug delivery device containing the patch.

For this reason, a fluid dynamics simulation was performed in ANSYS with a linear mesh of uniform elements and used 609,718 nodes and 592,845 elements. This simulation used a dose volume of 6 min and 0.4 mL (as in [49]), an input condition corresponding to the mass flow value, and an output condition with a magnitude of 0, reflecting the atmospheric pressure.

The mass flow of the MN patch was calculated with Equation (1). Here, it is necessary to know the volumetric flow (Q) and use Equation (2) to calculate the Q of a single MN.
(1)$$Dosis=\frac{0.4\;\mathrm{mL}}{6\;\min}=6.667\times 10^{-2}\;\frac{{\mathrm{cm}}^{3}}{\min}$$
(2)$$Q=\frac{6.667\times 10^{-2}\frac{{\mathrm{cm}}^{3}}{\min}}{25}=2.668\times 10^{-3}\;\frac{{\mathrm{cm}}^{3}}{\min}=4.447\times 10^{-5}\;\frac{{\mathrm{cm}}^{3}}{s}$$

The mass flow rate (m) was calculated using Equation (3) as input for the simulation. The density of insulin, 1.0036 $\frac{g}{{\mathrm{cm}}^{3}}$, was used [50], in addition to the viscosity of insulin, $1.1\;\frac{\mathrm{mPa}}{s}$ [33].
(3)m=Q×ρ. 
$$m=4.463\times 10^{-8}\;\frac{\mathrm{Kg}}{s}$$

### 2.4. Computer-Aided Design (CAD) of MN Patch

The MN patch design was performed with Solidworks (version 2020-2021 SP5.0, Dassault Syestems) software. The patch consists of 25 MNs (see Figure 2) organized in 5 columns and 5 rows, with dimensions 10 mm × 8 mm × 0.5 mm. The dimensions of the MNs in this work are within the range of dimensions used in MN design, which are lengths within 150 μm to 1500 μm and widths within 50 μm to 250 μm [20]. Different authors have previously used this number of microneedles in a patch for transdermal drug delivery and obtained promising results [12,13,15].

The beveled tip design ensures that the most prominent part of the MN will initially pierce the skin, allowing a micro-perforation that will gradually widen as the MNs pierce the skin, preventing the needle from clogging in the process. 

### 2.5. MN Patch Manufacturing Process

The 3D-printed transdermal patch was manufactured with Form2 SLA technology with a Class 1 resin evaluated according to ISO 10993-1. The MN patch was modeled in Solidworks and exported to PreForm, the 3D printing preparation software. In this software, we configured the printing angle, supports, raft, and layer thickness. The printing angles of the parts were −45°, 0°, 45°, and 90° concerning the build platform, i.e., the angle between the plane of the patch base and the plane of the build platform (see Figure 3). The supports were generated manually with a touchpoint size of 0.30 mm and a density of 1. The raft type was configured as a full raft with a thickness of 3 mm. Thinner layers reduce the stair-step effect achieving better print quality on the part’s surface. For this reason, the chosen thickness was the minimum allowed by the form 2 printer with surgical resin.

Once the microneedle patch was printed, it was immersed in 99% isopropyl alcohol at different time intervals (2, 4, 6, 8, and 10 min) to determine which time optimized the process (Figure 4B). For this, the process was repeated three times at different times; a 2 min wash exposure was not enough to remove excess resin. Then, it was placed in an ultrasonic cleaner for 10 min (Figure 4C); this is necessary because the patch finishes printing it with excess resin. Once the patch was cleaned with alcohol, it was left to dry for approximately 20 min before the curing process (UV radiation) in the FormCure for 30 min at 60 °C (see Figure 4D).

This work took advantage of the capability of 3D printing technologies to fabricate small structures in the order of micrometers to have an effective, reproducible, and accurate method. A total of eight patches were printed, and further study is expected to evaluate more details of the printed patches.

## 3. Results and Discussions

### 3.1. Structural Design of MN

Mechanical stimulation ensures a safety factor of 4, which is a good safety factor for future use in the medical field [51,52]. In addition, a compression force of 0.5 N [49] was used to simulate the force required to penetrate the skin. This force value is within the range of 0.1–3 N, sufficient to allow insertion on command, found by other authors [53].

These simulations showed Von Misses Stresses of 18.057 MPa (see Figure 5) and maximum deformation of $2.179\times 10^{-3}$ in the designed MN without presenting any irreversible adverse effect.

From the wash process applied to the 3D-printed MN, it can be observed that when exposed to 10 min of alcohol, the piece could be clean and was measured on a VR–3000 macroscope and compared with CAD geometry (Figure 6).

### 3.2. Flow Simulation 

Flow simulation was configured to obtain the pressure contours at the top and the flow velocity that the MN needs to achieve the desired characteristics. The MN mesh is made up of the internal volume of the microneedle, i.e., the volume through which the fluid will flow. The results of this simulation show that the pressure required at the top of the MN to ensure a Q magnitude of 4.447 × 10^−5^ $\frac{{\mathrm{cm}}^{3}}{s}$ was 1.084 Pa (see Figure 7) and a fluid velocity of 4.800 $\frac{m}{s}$ (see Figure 8) is needed inside the microneedle to maintain the defined Q. 

### 3.3. Manufacture of the Transdermal Patch

The patch was printed at different angles (−45°, 45°, 0°, and 90°) concerning the build platform. The best patch printing quality was obtained when the printing angle was 0° since, at this angle, a structure of the patch and microneedles very similar to that of the CAD model was obtained. Figure 9 shows the structure of some of the printed patches.

At the end of the printing process, all the microneedle internal diameters were clogged. However, 40% of these holes were easier to unclog after the cleaning process, and the remaining 60% required a drilling process with a 0.100 mm micro-milling in a router CNC to re-work each hole and improve the inner diameter, as seen in Figure 10.

Meanwhile, the fabrication process of the MN patch took approximately 2.5 h.

## 4. Future Perspectives

In future work, we will explore different resins and printing parameters to improve the fabrication of the patches and obtain the holes in the MNs. Additionally, we plan to review the penetration of the current design and explore shape variations to optimize the penetration of the MNs. In this regard, the SLA manufacturing of a One-Plane Bevel-Tipped MN has been recently reported [23]. Additionally, we will explore the behavior of fluid flow through the MN and connect the patch with a microelectromechanized system to provide microdosing.

## 5. Conclusions

An MN patch for transdermal drug delivery was manufactured with a biocompatible resin using a 3D printer by SLA. The obstructed internal holes were re-marked by making micro-perforations. The print quality was good, as the appearance of the patch closely matched the geometry of the CAD model, indicating that this method of microneedle manufacture has considerable contributions to device fabrication. However, further research is needed to investigate which printing parameters could improve each of the prints made, specifically to improve the print quality of the MN tip and the internal hole. 

The patch was made with 25 MN; the microneedles were validated with a mechanical simulation. The simulation confirmed the dimensions of this, and the material can withstand resistance to skin penetration during MN penetration. The simulation showed no deformation or irreversible damage to the structure when applying the necessary force to penetrate the skin. The MN could be coupled into a transdermal drug delivery method. These systems are typically applied to the upper arm or forearm [54,55].

In addition, a safety factor of 4 in the MN structure was maintained, which gives the maximum stress presented by the structure when subjected to the insertion force much lower than the tensile strength of the material, thus avoiding plastic deformation. 

Furthermore, with the flow simulation, it was possible to find the pressure required at the top of the MN and the velocity needed to maintain the initial parameters given. These parameters are of great importance for the future creation of the device to allow proper drug dosing. For this reason, the parameters found in this work are of vital importance and make a considerable contribution to transdermal drug delivery devices.

## Figures and Tables

**Figure 1 micromachines-13-02190-f001:**
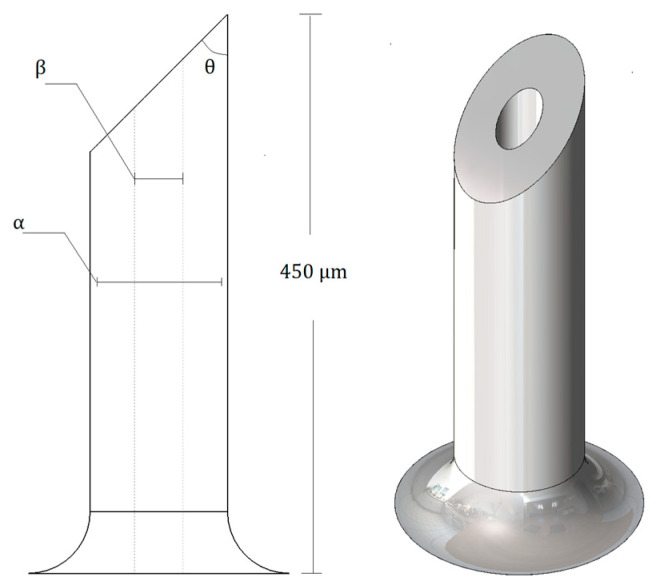
Microneedle CAD Model.

**Figure 2 micromachines-13-02190-f002:**
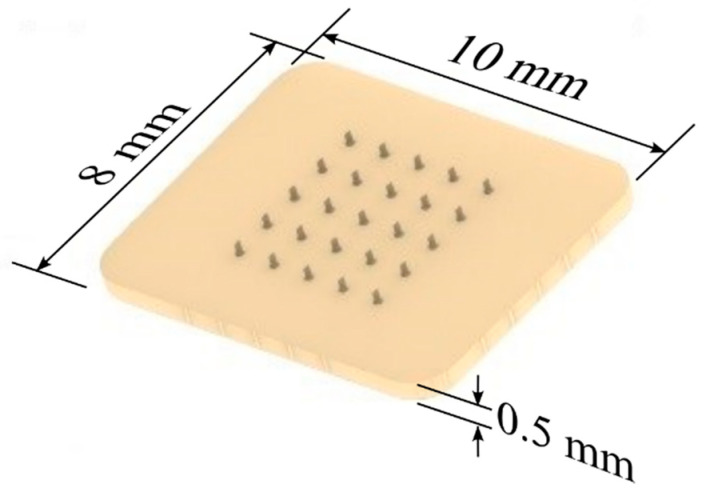
Rendering of the MN patch CAD model.

**Figure 3 micromachines-13-02190-f003:**
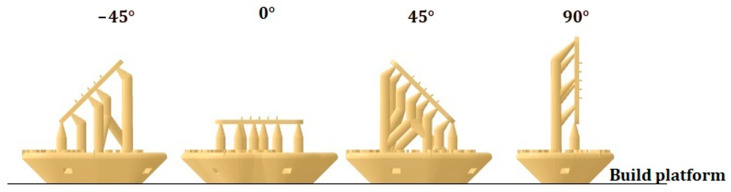
Printing angles concerning the build platform.

**Figure 4 micromachines-13-02190-f004:**
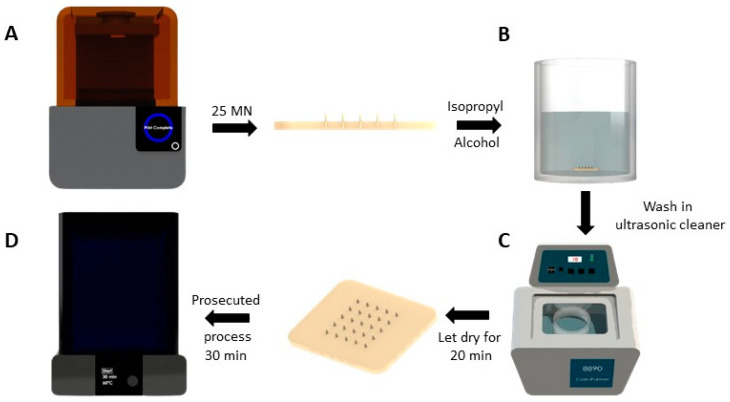
MN patch manufacturing process. (**A**) MN patch manufacturing in the Form 2 printer, (**B**) MN patch immersed in isopropyl alcohol, (**C**) ultrasonic cleaner to eliminate excess resin, (**D**) cured MN patch in FormCure at 60 °C.

**Figure 5 micromachines-13-02190-f005:**
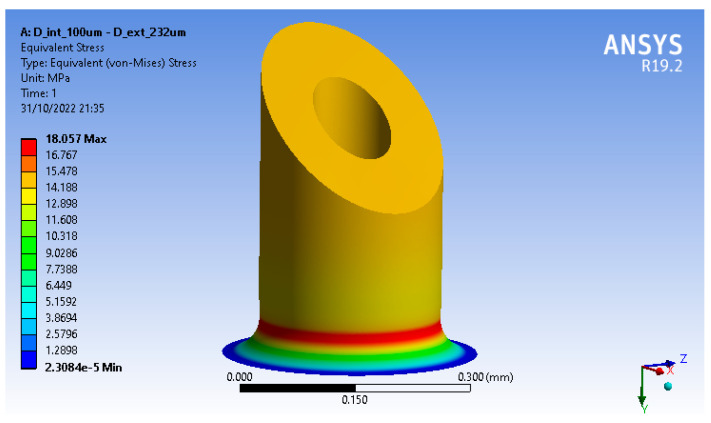
Von Misses Tensions of microneedle.

**Figure 6 micromachines-13-02190-f006:**
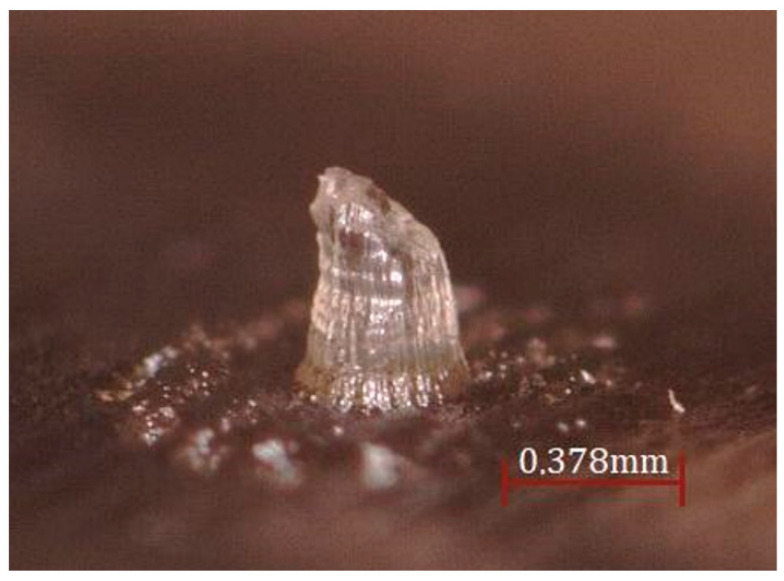
Patch MN.

**Figure 7 micromachines-13-02190-f007:**
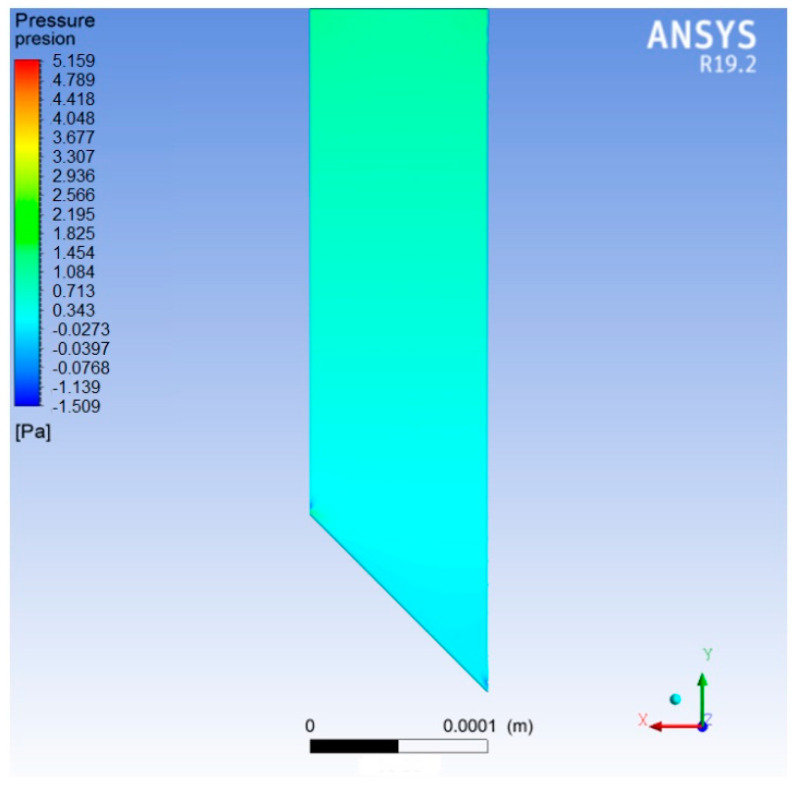
Contours of pressure for the microneedle.

**Figure 8 micromachines-13-02190-f008:**
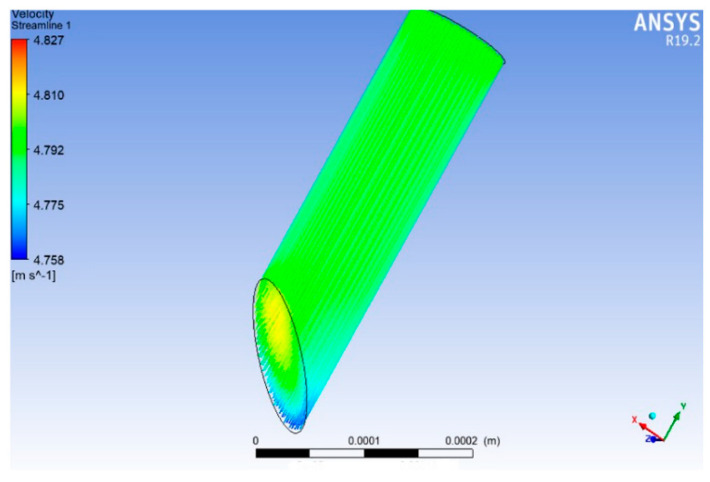
Velocity vectors for the microneedle outlet.

**Figure 9 micromachines-13-02190-f009:**
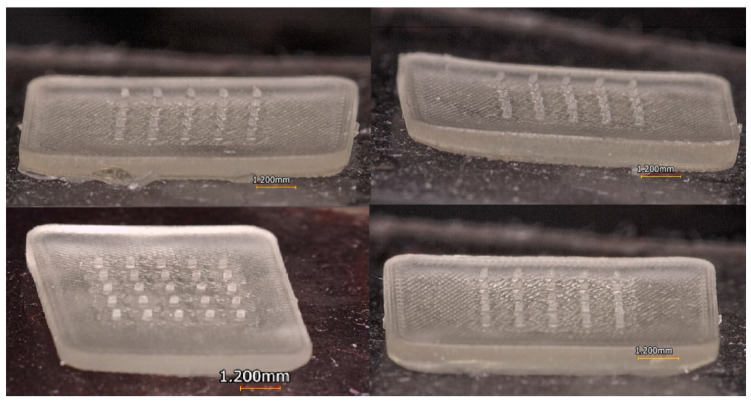
Manufacture of MN patches.

**Figure 10 micromachines-13-02190-f010:**
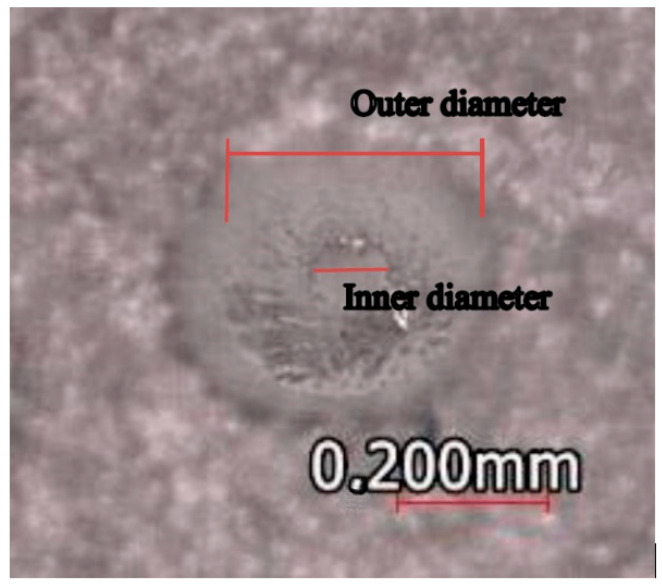
Microneedle internal diameter drilling—Top view.

## Data Availability

Not applicable.

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
