# Peer review of "Manufacturing of a Transdermal Patch in 3D Printing"

_micromachines, 2022, doi:10.3390/mi13122190_

Round 1

Reviewer 1 Report

In this manuscript, the authors adopted 3D stereolithography printing to prepare a transdermal patch composed of 25 microneedles. The finite element analysis was conducted to obtain the mechanical behavior of the printed microneedles.  I suggest the publication of this manuscript in Micromachines, a high-performance journal if the author can address the following concerns.

1. The author should use the same significant figures in this manuscript. For example, in the sentence “The values obtained through the analysis were: a Von Misses stress of 18.057 MPa, a maximum deformation of 0.0021792, and a safety factor of 4”, I do not think it was reasonable to have so many significant figures since the result changed as for every test.

2. In equations (1), (2), (3), and (4), the typeface should be the same as that in the main part of the manuscript. Despite this small error, the author should be careful since this reflected whether the author was well trained.

3. The author printed the microneedles using biocompatible resin. One recent paper (DOI: 10.1002/adfm.202107437) summarized the progress in the 3D printing of hydrogels. To give readers more background, the author can compare the similarity and differences between the printing of resins and hydrogels.

4. The author claimed that “Furthermore, the parameters found in this work are of great importance for the future implementation of a transdermal drug delivery device.”, however, the reviewer did not see any applications in the field of drug delivery. Besides, resins for drug delivery were not common. Please explain.

Author Response

Thank you for your time in helping us improve our manuscript. You are correct; for that reason, we have made the following corrections to the document

Reviewer 1

In this manuscript, the authors adopted 3D stereolithography printing to prepare a transdermal patch composed of 25 microneedles. The finite element analysis was conducted to obtain the mechanical behavior of the printed microneedles. I suggest the publication of this manuscript in Micromachines, a high-performance journal if the author can address the following concerns.

The author should use the same significant figures in this manuscript. For example, in the sentence “The values obtained through the analysis were: a Von Misses stress of 18.057 MPa, a maximum deformation of 0.0021792, and a safety factor of 4”, I do not think it was reasonable to have so many significant figures since the result changed as for every test.

Response: Dear reviewer, thank you for your comment, we review carefully the figure and left the important as requested. The number of decimal places in the values obtained was modified.

” to  “

4.8” to “ 4.800

” to “

Location: Abstract section

  1. In equations (1), (2), (3), and (4), the typeface should be the same as that in the main part of the manuscript. Despite this small error, the author should be careful since this reflected whether the author was well trained.

Response: The equations have a default typeface, for this reason we change the typeface of all numbers and their units to the same typeface that the equations have

Location: Equations (1), (2), (3), and (4)

  1. The author printed the microneedles using biocompatible resin. One recent paper (DOI: 10.1002/adfm.202107437) summarized the progress in the 3D printing of hydrogels. To give readers more background, the author can compare the similarity and differences between the printing of resins and hydrogels.

Response: The introduction was modified as suggested by the reviewer and corrected in the document. We have changed the seventh paragraph of the introduction to highlight the work published in 2021: the importance of 3D printing of modeling and simulations in the dimensions of micrometers.

In 2021, Ge et al. presented the different advantages and challenges that arise in micromanufacturing with 3D printing; in their study of hydrogels, they show the difficulties that occur with other materials when being manufactured at micrometer dimensions, but from their conclusions, they show the importance of computational studies using models and simulations that help to understand the manufacturing process better [21].”

Location: Section 1. Introduction, page 2.

  1. The author claimed that “Furthermore, the parameters found in this work are of great importance for the future implementation of a transdermal drug delivery device.”, however, the reviewer did not see any applications in the field of drug delivery. Besides, resins for drug delivery were not common. Please explain.

    Response: Thank you for your comment, a section named “Future perspectives” was added before conclusion section to attend your inquiry.

Gittard et al. [37] discusses "two-photon polymerization", a laser-based rapid prototyping technique, and its use in the fabrication of hollow and solid microneedles. It describes how this technique can be used to make microneedles with exceptional mechanical properties and discusses the potential applications of microneedles in drug delivery.

  1. Lu et al. [38] present the fabrication of drug-loaded microneedle arrays using microstereolithography. These arrays are made of poly (propylene fumarate), which is mixed with diethyl fumarate to control viscosity and improve mechanical properties. The arrays are able to release dacarbazine at a controlled rate for five weeks.

Dardano [39] reports on two procedures for fabrication of polymeric microneedles and shows how Microneedles from a hundred microns up to two millimeters can be obtained by changing the exposure time and/or the power density during photolithography. In addition, different microneedle shapes, such as cylindrical, conic or lancet-like, can be achieved for specific applications such as micro-indentation or drug delivery.

In [40] Doraiswamy et al., discusses the fabrication of microneedles using two photon polymerizations, and shows that these microneedles can be used for transdermal delivery of nanoscale pharmacologic agents.

Ali et al. [41] proposes a micro fabrication process for polymer micro needles, using DLP based projection-based stereo lithography. The fabrication is performed with continuous movement of the platform in the vertical direction. The results indicate that polymer micro needles with appropriate geometry can be fabricated using this technique.

Recently, Q. Yang et al. [42] mention the high precision and accuracy has enabled SLA technique to successfully fabricate the hollow microneedles, which is capable to transdermal deliver drug solution. Also, Xenikakis and co-workers [43] reported a typical solid transdermal microneedle array using a polymer-based material. They utilized finite element analysis (FEA) to simulate the insertion the printed microneedle arrays. Their in-vitro results indicated that those 3D-printed microneedle arrays possessed sufficient penetration ability on human skin, and also significantly promoted the transdermal transportation of the model dyes.

  1. Gittard, S.D.; Ovsianikov, A.; Chichkov, B.N.; Doraiswamy, A.; Narayan, R.J. Two-photon polymerization of microneedles for transdermal drug delivery. Expert Opin Drug Deliv. 2010; 7 (4), 513-533. https://doi.org/10.1517/17425241003628171.
  2. Lu, Y.; Mantha, S.N.; Crowder, D.C.; et al. Microstereolithography and characterization of poly (propylene fumarate)-based drug-loaded microneedle arrays. Biofabrication. 2015; 7 (4), 045001. Published 2015 Sep 29. https://doi.org/10.1088/1758-5090/7/4/045001.
  3. Dardano, P.; Caliò, A.; Di Palma, V.; Bevilacqua, M.F.; Di Matteo, A.; De Stefano, L. A Photolithographic Approach to Polymeric Microneedles Array Fabrication. Materials (Basel). 2015; 8 (12), 8661-8673. Published 2015 Dec 11. https://doi.org/10.3390/ma8125484.
  4. Doraiswamy, A.; Ovsianikov, A.; Gittard, S.D.; et al. Fabrication of microneedles using two photon polymerization for transdermal delivery of nanomaterials. J Nanosci Nanotechnol. 2010; 10 (10), 6305-6312. https://doi.org/10.1166/jnn.2010.2636.
  5. Ali, Z.; Türeyen, E. B.; Karpat, Y.; Çakmakcı, M. Fabrication of Polymer Micro Needles for Transdermal Drug Delivery System Using DLP Based Projection Stereo-lithography, Procedia CIRP, 2016; 42, 87–90, https://doi.org/10.1016/j.procir.2016.02.194.
  6. Yang, Q.; Zhong, W.; Xu, L.; et al. Recent progress of 3D-printed microneedles for transdermal drug delivery. Int J Pharm. 2021; 593, 120106. https://doi.org/10.1016/j.ijpharm.2020.120106.
  7. Xenikakis, I.; Tzimtzimis, M.; Tsongas, K.; et al. Fabrication and finite element analysis of stereolithographic 3D printed microneedles for transdermal delivery of model dyes across human skin in vitro. Eur J Pharm Sci. 2019; 137, 104976. https://doi.org/10.1016/j.ejps.2019.104976.

Reviewer 2 Report

This paper describes the design and manufacturing of a microneedle system to be assembled in transdermal patches for insulin administration. In this study the part is manufactured using additive manufacturing, and in particular the stereolitography process. The topic of this study is of potential interest for the bioengineering community, but the paper has to be significantly improved to be considered for publication in a journal.

Indeed, the reported state of the art is not very extensive and is not suitable to prove the paper contribution to the present knowledge. Moreover, the technical treatment is poor because many experimental details and results are missing (see below for more detailed comments).

The overall writing quality is good (there are just a few minor issues to be corrected, e.g. the use of the personal construction in some sentences of the abstract).

DETAILED REVIEW

1. Introduction

-      pag 1, lines 33 and 38: Define the acronyms.

-      pag 2, lines 56-58: This sentence should be moved after the sentence of lines 51-52.

-      pag 2, lines 59-60: You should list some examples of the used micro-modeling techniques and add some related references.

-      pag 2, lines 64-67:

o  Do microscale needle dimensions represent a challenge for 3D printing?

o  What are the problems in printing quality that can be caused by the sensitivity to different pre-printing and post-printing parameters?

-      To improve this section you have to:

§ explain the reasons for selecting the stereolitography as manufacturing process;

§ cite and discuss a number of existing papers describing the manufacturing of microneedles by stereolitography;

§ highlight the original contribution of this paper with respect to the existing literature.

2. Materials and Methods

2.2. Computer aided design (CAD) of MN

-      pag 2, lines 86-89: Is the selection of microneedle geometry and dimensions described in detail in [23]?

-      pag 3, lines 92-93: The sentence is not complete.

2.2. Finite element analysis of MN

-      The section number has to be “2.3”.

2.2. Computer aided design (CAD) of MNs patch

-      The section number has to be “2.4”.

-      pag 4, lines 122-123: What is the patch thickness?

2.2. MMs MNs patch manufacturing process

-      The section number has to be “2.5”.

-      pag 4, lines 132-133:

o  Define the printing angle.

o  Add further details on the printing settings, e.g. use of raft, use of supports, layer thickness, etc.

-      pag 4, lines 133-135: Is it really possible to select the minimum resolution / laser spot size?

3. Results and Discussions

3.3. Manufacture of the transdermal patch

-      pag 6, lines 169-176: These contents are more appropriate to Section “MMs patch manufacturing process”.

-      pag 6, lines 177-178: Which alternative settings have been tested and have given a lower quality?

-      pag 6, lines 178-184: These contents are more appropriate to Section 1.

-      pag 7, lines 185-187: These contents are more appropriate to Section “Computer aided design (CAD) of MN”.

-      Add the overall manufacturing time.

-      Have you printed just one patch?

5. Conclusions

-      The section number has to be “4”.

-      pag 7, lines 194-195:

o  How many holes out of 25 microneedles were obstructed?

o  Add more details about the hole re-manufacturing.

-      pag 7, lines 195-197: Have you performed some measurements of the printed patch (patches) or have you compared it (them) with the CAD model only “by eye”?

-      pag 7, lines 197-200: Have you tested the printed patch (patches) e.g. for fluid flow or skin penetration?

Author Response

Thank you for your time in helping us improve our manuscript. You are correct; for that reason, we have made the following corrections to the document

Reviewer 2

This paper describes the design and manufacturing of a microneedle system to be assembled in transdermal patches for insulin administration. In this study the part is manufactured using additive manufacturing, and in particular the stereolitography process. The topic of this study is of potential interest for the bioengineering community, but the paper has to be significantly improved to be considered for publication in a journal.

Indeed, the reported state of the art is not very extensive and is not suitable to prove the paper contribution to the present knowledge. Moreover, the technical treatment is poor because many experimental details and results are missing (see below for more detailed comments).

The overall writing quality is good (there are just a few minor issues to be corrected, e.g. the use of the personal construction in some sentences of the abstract).

DETAILED REVIEW

  1. Introduction

-      pag 1, lines 33 and 38: Define the acronyms.

Response: The acronyms were defined

World Health Organization (WHO)”

Pan American Health Organization (PAHO)”

Location: Introduction section, page 2

-      pag 2, lines 56-58: This sentence should be moved after the sentence of lines 51-52.

Response: The sentence: “MNs have lengths from 150 μm to 1500 μm, this being a sufficient length to release the drug into the epidermis and the width of an MN can be between 50 μm and 250 μm [19].” have been moved after paragraph “Systems composed of microneedles (MN) assembled in a transdermal patch provide a unique route of administration [1], innovative and with promising results [11-19].”. as requested.

Location: Section 2.2., page 3

-      pag 2, lines 59-60: You should list some examples of the used micro-modeling techniques and add some related references.

Response: Some examples of the micro-modeling techniques are given as requested:

However, other applications of the micro-modeling approaches can be the 3D reconstruction of biological tissues [20, 21], numerical simulation (e.g., recovery of gas hydrates) [22], computational platform (e.g. finite element for heart ventricles) [23], finite element analysis of rubberized concrete interlocking masonry [24], amongst others.

Location: Introduction section, page 2

-      pag 2, lines 64-67:

o  Do microscale needle dimensions represent a challenge for 3D printing?

o  What are the problems in printing quality that can be caused by the sensitivity to different pre-printing and post-printing parameters?

-      To improve this section you have to:

  • explain the reasons for selecting the stereolitography as manufacturing process;
  • cite and discuss a number of existing papers describing the manufacturing of microneedles by stereolitography;
  • highlight the original contribution of this paper with respect to the existing literature.

Response: The introduction was modified as suggested by the reviewer and corrected in the document. We have changed the seventh paragraph of the introduction.

On the other side, this MN patch fabrication technique is a valuable contribution to microneedle manufacture techniques since conventional techniques such as micro-milling, and injection molding, among others, have some cost-saving limitations [20]. While, Stereolithography (SLA) printed patches present important characteristics such as low cost of printers, printing inks, and fast manufacturing times [13] that will allow this technique to make considerable contributions in the manufacture of devices for transdermal drug delivery. In 2021, Ge et al. presented the different advantages and challenges that arise in micromanufacturing with 3D printing; in their study of hydrogels, they show the difficulties that occur with other materials when being manufactured at micrometer dimensions, but from their conclusions, they show the importance of computational studies using models and simulations that help to understand the manufacturing process better [21].”

Location: section 1. Introduction.

  1. Materials and Methods

2.2. Computer aided design (CAD) of MN

-      pag 2, lines 86-89: Is the selection of microneedle geometry and dimensions described in detail in [23]?

Response: First, reference [23] has been changed, and now it is reference [30]. On the other hand, reference [30] describes the selection of the geometry and dimensions of the microneedles in detail.

Location: Reference [30]

-      pag 3, lines 92-93: The sentence is not complete.

Response:  The sentence was completed, thank you for the observation.

Previous sentence

“This relatively simple MN structure allows an adequate load distribution over the structure, allowing the penetration of the MN into the skin without structural dam”.

Complete sentence

“This relatively simple MN structure allows an adequate load distribution over the structure, allowing the penetration of the MN into the skin without structural damage”.

Location: Section 2.2, page 4

2.2. Finite element analysis of MN

-      The section number has to be “2.3”.

Response: Ready, thanks for the observation

Location: Subsection 2.3, “Finite element analysis of MN”

2.2. Computer aided design (CAD) of MNs patch

-      The section number has to be “2.4”.

Response: Ready, thanks for the observation

Location: Subsection 2.4, “Computer aided design (CAD) of MNs patch”

-      pag 4, lines 122-123: What is the patch thickness?

Response: The patch thickness was added in the following text

“The MN patch design was performed in Solidworks 2019 software. The patch consists of an array of 5 x 5 MNs (see fig 2) and its dimensions of 10 mm x 8 mm x 0.5 mm”

We also changed Figure 2 and we added the thickness of the patch to it.

Location: Section 2.4, page 4

2.2. MMs MNs patch manufacturing process

-      The section number has to be “2.5”.

Response: Ready, thanks for the observation

Location: Subsection 2.5, “MNs patch manufacturing process”

-      pag 4, lines 132-133:

o  Define the printing angle.

o  Add further details on the printing settings, e.g. use of raft, use of supports, layer thickness, etc.

Response: The following text was modified and the requested information was added

Previous text

 “In this software the printing angle of the parts was configured to be 0° concerning the printing platform. Then the print file was sent to the Form 2 printer and a minimum resolution of 50 µm was configured, which is given by the size of the focal point of the laser

Modified text

 “In this software, we configured the printing angle, the supports, the raft, and the layer thickness. The printing angle of the parts is 0° concerning the printing platform. The supports were generated manually with a touchpoint size of 0.30 mm and a density of 1. The raft type was configured as a full raft with a thickness of 3 mm. A minimum layer thickness of 50 µm was configured. The minimum value of layer thickness depends on the printer and resin selected. Thinner layers capture fine details achieving better print quality and resolution printing. For this reason, in the present work chosen minimum layer thickness value is allowed by the form 2 printer and the surgical resin

Location: Section 2.5, page 5

-      pag 4, lines 133-135: Is it really possible to select the minimum resolution / laser spot size?

Response: What is selected is the layer thickness. The layer thickness affects both the speed and the quality of the print. Thicker layers allow faster printing but sacrifice detail. Thinner layers allow better detail and print resolution but slower printing.

The minimum value of the layer thickness depends on the printer and the resin used.

To clarify this doubt in the text a paragraph is morphed into the text

Previous text

Then the print file was sent to the Form 2 printer and a minimum resolution of 50 µm was configured, which is given by the size of the focal point of the laser

Modified text

A minimum layer thickness of 50 µm was configured. The minimum value of layer thickness depends on the printer and resin selected. Thinner layers capture fine details achieving better print quality and resolution printing. For this reason, in the present work chosen minimum layer thickness value is allowed by the form 2 printer and the surgical resin

Location: Section 2.5, page 5

  1. Results and Discussions

3.3 Manufacture of the transdermal patch

-      pag 6, lines 169-176: These contents are more appropriate to Section “MMs patch manufacturing process”.

Response: The following text was relocated to section 2.5 as requested

Manufacture of the 3D printed transdermal patch was performed in Form2 stereo-lithography (SLA) technology with a Class 1 resin that has been evaluated according to ISO 10993-1. All patches were washed in 99% isopropyl alcohol and then cured under UV radiation in the FormCure at 60°C for 30 minutes, parameters given by the manufacturer.

This work took advantage of the capability of 3D printing technologies to fabricate small structures in the order of micrometers to have an effective reproducible and accurate method.

Location: section 2.5

-      pag 6, lines 177-178: Which alternative settings have been tested and have given a lower quality?

Response: We printed the patch at different printing angles (-45°, 0°, 45° and 90°). Low print quality was obtained at the -45°, 0° and 45° angles. While the best print quality was obtained at the 0° angle.

This was added in the following text:

“The patch was printed at different angles (-45°, 45°, 0°, and 90°) concerning the printing platform. The best patch printing quality was obtained when the printing angle was 0° “

Location: Section 3.3, page 8

-      pag 6, lines 178-184: These contents are more appropriate to Section 1.

Response: The text was moved to section 1 as requested

On the other side, this MN patch fabrication technique is a valuable contribution to microneedle manufacture techniques since conventional techniques such as micro-milling, and injection molding, among others, have some cost-saving limitations [20]. While, Stereolithography (SLA) printed patches present important characteristics such as low cost of printers, printing inks, and fast manufacturing times [13] that will allow this technique to make considerable contributions in the manufacture of devices for transdermal drug delivery.

Location: Section 1, Introduction, page 2

-      pag 7, lines 185-187: These contents are more appropriate to Section “Computer aided design (CAD) of MN”.

Response: The following text was moved to the mentioned section as requested

Also, the beveled tip design ensures that the most prominent part of the MN will initially pierce the skin, allowing a microperforation that will gradually widen as the MN pierce the skin, preventing the needle from clogging in the process.

Location: Section 2.4, page 5

-      Add the overall manufacturing time.

Response: We added the overall time of manufacturing process in the following text:

“On the other hand, the manufacturing process of the MNs patch took approximately 2.5 hours.”

Location: Section 3.3., page 8

-      Have you printed just one patch?

Response: No, we were printed about 8 patches and further study is expected to evaluate more details of the printed patches.

  1. Conclusions

-      The section number has to be “4”.

Response: Ready, thanks for the observation, conclusion section was numbered as 5, because we added a new section “4. Future Perspectives”

Location: Section 5, “Conclusions”

-      pag 7, lines 194-195:

o How many holes out of 25 microneedles were obstructed?

Response: Several microneedles were initially obstructed after improving the process, as explained in section 3.1. the process was improved.

Location: section 3.1. Structural design of MN

o Add more details about the hole re-manufacturing.

Response: The Structural design of MN was modified as suggested by the reviewer and corrected in the document. We have included the last paragraph of the Structural design of MN.

Before subjecting the MNs to the curing process, the manufactured MNs should be washed to remove the remaining resin pieces. For this purpose, the MNs were soaked in 99% alcohol at different time intervals (2 min, 4 min, 6 min, 8 min, and 10 min) to observe which time was suitable for this process to optimize it. For this, the process was repeated three times at different times, where it was obtained that a 2 min wash exposure was not enough since the MN still had excess resin. When the MN was exposed to 10 min of alcohol, the piece could be seen as clean and compared with the simulation, see Figure 5.”

Location: section 3.1. Structural design of MN

-      pag 7, lines 195-197: Have you performed some measurements of the printed patch (patches) or have you compared it (them) with the CAD model only “by eye”?

Response: Yes, it was compared only with the simulation; no comparison was made with other models.

Location: section 3.1. Structural design of MN

-      pag 7, lines 197-200: Have you tested the printed patch (patches) e.g. for fluid flow or skin penetration?

Response: No, these tests will be carried out in future work.

Round 2

Reviewer 2 Report

See the attachment for comments.

Author Response

Dear Reviewer

Thanks for all your valuable comments, hope this time our corrections are pertinent and enough to improve the quality of this paper 

Response to the reviewer.

Introduction section, page 2

Please refer to the document (red = previous answers, green = our new answers)

Introduction

  • Reviewer

Based on the provided examples, it does not seem that the “micro-modeling techniques” can be used to physically fabricate the patches, as can instead be done by 3D printing. You have to clarify the whole paragraph.

  • Answer

The first sentence of the paragraph was rewritten (lines 25-26 on pag3):

  • Reviewer

              Your amendment has not provided satisfactory answers to the following points:

  • Do microscale needle dimensions represent a challenge for 3D printing? (comment on this topic)
  • Answer
  • We include texts that answer the question in the following lines: (lines 31-35 on pag 2) ;

  • Reviewer
    • what are the problems in printing quality that can be caused by the sensitivity to different pre-printing and post-printing parameters?
  • Answer
  • Examples of pre-printing and post-printing parameters that affect the printing quality were added for better understanding of that sentence (lines 41-43 pag3).

  • Reviewer
    • explain the reasons for selecting the stereolithography as manufacturing process (in particular, the reasons for selecting it among the other 3D printing techniques); highlight the original contribution of this paper with respect to the existing literature
  • Answer

Lines 33-38 on pag 3 were added to address this question

  • Reviewer

Some existing papers describing the manufacturing of microneedles by stereolithography have been cited and discussed in the new Section 4. However, that content should be part of Section 1 since it allows to describe the state-of-the-art and, then, to highlight the original contribution of your paper with respect to the existing literature.

  • Answer

In response to your request, we rewrite the information in the introduction, in the                   following lines:  sentence (lines 43-47 pag 2) ; (lines 1-13 pag 3) ; (lines 20-24 pag 3)

  1. Materials and methods

2.2. Computer aided design (CAD) of MN

  • Reviewer

You should cite the reference in Section 2.2.

  • Answer

Lines 24-25-on page 4 were added to address this suggestion

2.4. Computer aided design (CAD) of MNs patch

  • Reviewer

The figure now is clear, but the new sentence “The patch consists of 5 x 5 MNs (see fig 2), with dimensions of 10 mm x 8 mm x 0.5 mm.”is written in such a way that it seems that the dimensions are related tothe5 x 5 MNs.

  • Answer

The lines 31-33 on page 5 were rewritten.

2.5. MNs patch manufacturing process

  • Reviewer

The angle has not yet been defined: what are the two lines/planes between which the angle lies?

  • Answer

The lines 11-13 on page 6 were rewritten. Also, we have added figure 3 to better illustrate the printing angles.

  • Reviewer

I do not fully agree with your answer. A lower thickness does not improve the resolution in the plane parallel to the printing table and does not allow for producing finer details, but it rather produces a better quality of surfaces that are not parallel to the table, because it reduces the stair-step effect.

  • Answer

The lines 15-16 on page 6 were rewritten.

  1. Results and Discussions

section 3.1. Structural design of MN

  • Reviewer

If you add this content, you have to edit Section 2.5 to make it consistent.

  • Answer

Lines 11-13 on page 6, (section2.5) were added.

  • Reviewer

What was compared only with the simulation? Have you acquired 2D/3D digital images of        the patches?

  • Answer

The 3D printed model was compared with the CAD geometry. For this, a microscope VR–3000 measuring macroscope was used to measure dimensions on  MN. This process was added to lines 18-20 on page 7, section 3.1  

3.3. Manufacture of the transdermal patch

  • Reviewer

If you add this sentence, you have to edit Section 2.5 to make it consistent.

  • Answer

Line 11 on page 6 (section 2.5) was changed.

  • Reviewer

You should add this explanation to the paper and perhaps also some pictures of other patches, in order to support the robustness of the developed procedure.

  • Answer

lines 9-10 on page 9 and figure 9 on page 10 were added.

  1. Future perspectives
  • Reviewer

The content of this section is more appropriate to Section 1 since it depicts the state-of-the-art of microneedle manufacturing by stereolithography and does not draw future perspective on this topic.

  • Answer

The section 4 was rewritten

  1. Conclusions

  • Reviewer

Based on the sentence in Section 5, it seems that obstructed holes were present in the final version of the patch and had therefore been reworked. How many holes were obstructed? 

  • Answer

                Lines 11-14- page 9 were added to explain that (section 3.3)

  • Reviewer

If you add this content, you have to edit Section 2.5 to make it consistent.

However, this paragraph does not answer my comment and does not describe the re-working of holes(micro-drilling?). 

  • Answer

              Lines 12-14 describe in general the process of re-work hole (section 3.3)
